# Trace Element Concentrations in Degenerative Lumbar Spine Tissues: Insights into Oxidative Stress

**DOI:** 10.3390/antiox14040485

**Published:** 2025-04-17

**Authors:** Mikołaj Dąbrowski, Wojciech Łabędź, Łukasz Kubaszewski, Marta K. Walczak, Anetta Zioła-Frankowska, Marcin Frankowski

**Affiliations:** 1Adult Spine Orthopaedics Department, Poznan University of Medical Sciences, 61-545 Poznan, Poland; wlabedz23@gmail.com (W.Ł.); pismiennictwo1@gmail.com (Ł.K.); 2Department of Internal Medicine and Metabolic Disorders, Poznan University of Medical Sciences, Przybyszewskiego St. 49, 60-355 Poznan, Poland; martawalczak@ump.edu.pl; 3Analytical Chemistry, Faculty of Chemistry, Adam Mickiewicz University in Poznan, 61-614 Poznan, Poland; anetta.ziola-frankowska@amu.edu.pl; 4Department of Analytical and Environmental Chemistry, Faculty of Chemistry, Adam Mickiewicz University in Poznan, 61-614 Poznan, Poland; marcin.frankowski@amu.edu.pl

**Keywords:** trace elements, microelements, spine tissue, ICP-OES, intervertebral disc, degenerative disc disease, degenerative spine disease, oxidative elements

## Abstract

Degenerative changes are characterized by the formation of vertebral osteophytes, the hypertrophy of facet joints, and narrowing of the intervertebral space. This study aimed to investigate the concentrations of trace elements (Al, As, Se, Zn, Fe, Mo, Cu) in spinal tissues (intervertebral discs, muscle, and bone) of patients with degenerative lumbar spine disease (DLSD) and their potential associations with the disease. The research involved 13 patients undergoing surgery for symptomatic degenerative spine disease. The trace element concentrations were analyzed using chemical and radiographic assessments, with a statistical analysis performed through a Mann–Whitney U-test, Spearman’s rank correlation test, principal component analysis (PCA), and canonical discriminant analysis (CDA). The results showed significant variations and correlations among the trace elements across different spinal tissues, suggesting their roles in metabolic and oxidative processes and the pathology of spinal degeneration.

## 1. Introduction

Degenerative disc disease is a significant problem in developed countries [1,2]. Damage to the intervertebral disc leads to deterioration of the biomechanical properties of the motion segment and contributes to secondary processes in the joint capsule and ligamentous system, as well as hypertrophy of the elements of the posterior spinal column [3]. Degenerative changes are characterized by the formation of vertebral osteophytes, the hypertrophy of facet joints, and narrowing of the intervertebral space [4].

A quantitative analysis of the trace element content of human spine tissues, namely intervertebral discs (IVDs), muscle, and bone, is important to identify specific enzymes or metabolites that may be related to human degenerative lumbar spine disease (DLSD). In addition, it will allow the identification of metals that can significantly influence the metabolic progression in specific degenerative tissues of the spine.

Oxidative stress disrupts cellular homeostasis by increasing reactive oxygen species (ROS), which trigger proinflammatory cytokine release and degrade extracellular matrix components such as collagen. Reactive oxygen species (ROS) and reactive nitrogen species (RNS) have been shown to regulate matrix metabolism; promote a proinflammatory phenotype; and induce apoptosis, autophagy, and aging in intervertebral disc cells [5].

Trace elements, as essential cofactors in enzymatic reactions and redox processes, may contribute to the metabolic imbalance associated with spinal degeneration. Antioxidant trace elements such as zinc (Zn), copper (Cu), and selenium (Se) help modulate inflammatory pathways and protect against oxidative damage, both of which contribute to degenerative spinal diseases [6].

A quantitative analysis of trace elements in spinal tissues—intervertebral discs (IVDs), muscle, and bone—can provide insights into their involvement in enzymatic activity, oxidative damage, and disease progression. For example, studies have shown that assessing concentrations of elements such as iron (Fe), Zn, magnesium (Mg), and potassium (K) can be helpful in predicting the onset of degenerative changes in the spine [7].

Furthermore, identifying metals that significantly influence metabolic pathways may contribute to understanding the mechanisms underlying DLSD. The disruption of cellular mechanisms induced by trace elements can lead to the production of reactive oxygen species, causing oxidative stress and contributing to neuronal maladies. The concentrations of these trace elements were analyzed in the intervertebral disc tissue of patients undergoing surgery due to degenerative changes, and their correlation and similarities with other elements, including Mg, Zn, and Cu, related to tissue metabolism and degenerative processes were investigated in IVDs [3,4,8,9,10,11] and femoral bone [12,13,14,15,16].

The ongoing research primarily aims to conduct a comprehensive comparison of the microelement values found in diverse degenerative spinal tissues, with the overarching goal of shedding light on the intricate relationships and potential implications of these microelements in the context of spinal degeneration to elucidate their potential role in oxidative-stress-related metabolic pathways and spinal degeneration. The aim of the study was to comprehensively investigate the concentrations of several trace elements, including aluminum (Al), arsenic (As), selenium (Se), zinc (Zn), iron (Fe), molybdenum (Mo), and copper (Cu), in different spine tissues and determine their potential association with degenerative spine disease.

## 2. Materials and Methods

### 2.1. Patients

The study consisted of samples from 13 patients (3 women and 10 men, mean age of 58.77 ± 10.48) with symptomatic degenerative spine disease in the lumbar spine.

There were 5 cases with grade 2 osteophytes, 6 cases with grade 3 osteophytes, and 2 cases with grade 4 osteophytes in the facet joints. The Pathria scale scores were 8 for 2 cases and 5 for 3 cases. There was 1 case with grade 1, 7 cases with grade 2, and 5 cases with grade 3 scores according to the Goutallier scale. There were 5 cases with grade 2 and 8 cases with grade 3 scores according to the Modic scale. There were 3 cases with grade 2, 4 cases with grade 3, 3 cases with grade 4, and 3 cases with grade 5 scores based on the Pfirrmann scale (Table 1).

The qualification for the surgery involved presentation of the clinical symptoms of osteoarthritis of the spine in the form of local pain, radicular pain, and other neurological disorders (one and multi-level isolated DDD, stenosis of the spinal canal due to degenerative disease, and degenerative scoliosis). Before qualification for surgical treatment, the patients underwent at least six months of conservative treatment in the form of physiotherapy and pharmacotherapy. The Modic scale was predominantly represented by type II and III changes.

The tissues samples were collected during interbody fusion of the spine and perpendicular stabilization of the lumbar spine. In all procedures, removal of spinal tissues of paraspinal muscle (M), IVDs (D), the facet (F), and the spinous process (S) was performed, which is an integral part of the procedure that does not cause additional load or damage to the patient. We conducted a detailed interview with each patient, during which we controlled for the influence of diet, the environment, and occupational exposure on the study outcomes.

In addition to the laboratory analyses, pain and functional statuses were assessed using standardized scales. The Visual Analog Scale (VAS) was used to quantify the intensity of the pain, with higher scores indicating more severe pain. The Comi Back Scale was used to evaluate specific symptoms related to spinal disorders, and the Oswestry Disability Index (ODI) was used to assess the level of functional disability caused by back pain. These clinical measurements were incorporated into the analysis to explore the relationships between trace elements in the blood and spinal tissues with pain intensity and disability.

### 2.2. Chemical Analysis

#### 2.2.1. Preparation of Tissues and Blood Samples

The tissue samples were frozen at −40 °C after collection. Then, the articular process tissues were lyophilized for 24 h to constant mass in a Lyovac GT2e lyophilizer (Steris, Hamburg, Germany) to remove all residual water and after drying. The next step was related to the mineralization of the lyophilized samples, which included the addition of 8 mL of HNO_3_ (70%, purified by redistillation, ≥99.999% trace metals basis). In order to reduce analyte losses during further stages of sample preparation, after the addition of nitric acid, the samples were left for at least 8 h for a slow mineralization process. Then, the digestion was continued in a closed system using a high-pressure microwave oven (Microwave Digestion System, Anton Paar Multiwave PRO, Graz, Austria) equipped with an 8NXF 100 rotor. As part of the optimization process, the microwave energy, microwave energy increase time, microwave energy holding time, and cooling time were selected accordingly. The optimal values were set to 15 min to increase the microwave energy to 1500 W, then 30 min for the holding time and cooling time at a vessel temperature of 70 °C. After this time, the samples were transferred volumetrically to Falcon tubes. The samples obtained after digestion were clear, colorless, and without visible sediments or fat residues. In order to ensure quality control, blank samples were mineralized with each batch of samples.

The blood samples were stored in the fridge after collection. The next step was related to mineralization, which included the addition to each blood sample (100 µL) of 300 µL of HNO_3_ (70%, purified by redistillation, ≥99.999% trace metals basis), 100 µL of HCl (30%, purified by redistillation, ≥99.999% trace metals basis), and 100 µL of H_2_O_2_ (35% extra pure). Then, the sample after mineralization was diluted with double-distilled water for a further analysis using the ICP-MS technique.

#### 2.2.2. Reagents

For all solutions, demineralized water (sourced from the Milli-Q Direct 8 Water Purification System, Millipore, New York, NY, USA) was used for the experimental procedures. Additionally, the 70% nitric acid used and 30% HCl, which had a purity of Suprapure^®^, were of spectroscopic grade and obtained from Merck, USA. Hydrogen peroxide (35% by weight), of extra pure grade (Honeywell, 95299, Charlotte, NC, USA), was also used. Additionally, to eliminate any potential risk of metal contamination throughout the analysis and preparation steps, we subjected all plastic and glassware used, along with other equipment, to apply a cleaning process with immersion in 0.5 mol/L of nitric acid for a duration of 24 h. Standard solutions for the investigated trace elements (27Al; 75As; 63Cu; 56Fe; 98 Mo; 77Se; 80Se; 66Zn) were prepared by diluting certified solutions to a concentration of 1000 mg L^−1^, traceable to the Merck Standard Reference Material (New York, NY, USA). The reference materials for calibration (as internal standards) were 45Sc, 103Rh, and 9Be from Merck (Merck Millipore, New York, NY, USA).

#### 2.2.3. Determination of Trace Elements by ICP-MS Technique

The methodology for determining the trace elements in degenerative spine tissues and blood samples was an ICP-MS analytical technique because of its precision and capability for simultaneous multielement analyses. The analysis of Al, As, Cu, Fe, Mo, Se, and Zn was performed using an ICP-MS 2030 (Shimadzu, Kyoto, Japan) equipped with Collision Cell Technology. The instrumental parameters set for the analysis, including the plasma power, nebulizer flow rate, and reaction cell conditions, were optimized to achieve the best performance for the analysis of Al, As, Cu, Fe, Mo, Se, and Zn. For the trace element analysis, the concentration ranges of the standards were selected to cover the expected levels of these metals in spine tissues to ensure accurate calibration and quantification over a wide concentration spectrum. We obtained high correlation coefficients (R) for all investigated trace elements (R = 0.999 for Al, R = 0.999 for As, R = 0.999 for Cu, R = 0.998 for Fe, R = 0. 999 for Mo, R = 0.999 for Se, and R = 0.99980 for Zn), which confirmed the linearity, precision, and accuracy of the ICP-MS measurements.

A data analysis was conducted by comparing the intensity of the ions detected for each trace element with calibration curves determined using standard solutions of each known concentration. This comparison allowed for the quantification of the trace elements in the various types of degenerative spine tissues. All procedural details, including the specific settings used for the ICP-MS analysis, were thoroughly documented.

### 2.3. Radiographic Assessment

The Pfirrmann scale grades for intervertebral disc degeneration were assessed on MRI T2-weighted images [17]. The Modic scale is a classification system describing MRI signal changes in vertebral endplates associated with spinal degeneration [18]. The Goutallier classification is a visual assessment method used to evaluate the fatty infiltration of paraspinal muscles. Quantitative measurements obtained through MRI can provide more precise information about the muscle composition and extent of infiltration. While fatty infiltration and muscle morphology have been correlated with anthropometric parameters, their association with patient-reported pain or disability in lumbar spinal stenosis is still unclear [19]. The Pathria scale evaluates lumbar spine degeneration. The suggested standards are 3–5 grades, starting with “grade 0” for the non-degenerated state [20]. The grade of facet joint osteophyte was classified using Grogan’s methods [21].

### 2.4. Statistical Analysis

The analysis used Statistica software (Version 13.0, StatSoft Inc., Tulsa, OK, USA). To compare the impacts of various environmental factors on the concentration of analyzed metals in the bone, we used a Mann–Whitney U-test (*p* < 0.05). In addition, we determined the Spearman’s rank correlations between the trace elements occurring in the materials and between the different studied trace elements in different tissues.

A chemometric analysis was performed to evaluate variables from the independent assumption showing the mutual relationships between the analyzed factors by applying a principal component analysis (PCA) and canonical discriminant analysis (CDA). A CDA is a multivariate technique that can be used to determine the relationships among a categorical variable (tissues) and a group of independent variables (trace elements). Biplots of the canonical discriminant analysis were performed to visually analyze the influences of taxonomy on the multielement composition of tissues. The canonical discriminant analyses were conducted using JMP Pro 18 (Sas, Cary, NC, USA).

### 2.5. Ethics Statement

The use of IVD samples in this research was permitted by the Bioethical Committee of the University of Medical Sciences in Poznan (Poland) (permit no. 444/20; date: 17 July 2020), and all patients provided written informed consent prior to participation.

## 3. Results

The highest mean concentration of aluminum can be found in the paraspinal muscle, followed by the IVD, spinous process, and facet. The interquartile ranges indicate the variability in aluminum levels across the tissues.

Cu shows relatively low concentrations in all tissues, with the paraspinal muscle having the highest mean concentration. The values for Cu are relatively consistent across the tissues, as indicated by the narrow interquartile ranges.

The As levels are highest in the paraspinal muscle and IVD. The facet and spinous process show relatively lower mean concentrations of arsenic. The IQR values suggest some variability in As levels, especially in the paraspinal muscle.

The Mo concentrations are generally low across all tissues, with the paraspinal muscle having the highest mean concentration. The facet and spinous process show similar mean values, while the IVD has a slightly higher mean concentration.

The Se levels are relatively consistent across the tissues, with the paraspinal muscle showing the highest mean concentration. The interquartile ranges indicate a moderate amount of variability in Se levels.

The Zn concentrations are highest in the paraspinal muscle, followed by the spinous process, facet, and IVD. The paraspinal muscle exhibits the widest interquartile range, indicating some variability in Zn levels.

Fe shows the highest mean concentration in the paraspinal muscle, followed by the IVD, facet, and spinous process. The interquartile ranges suggest variability in iron levels, particularly in the paraspinal muscle (Table 2, Figure 1).

Here, the paraspinal muscle displayed the highest median values in terms of the content of the trace metals under investigation. It is worth noting that in the case of the IVD, Fe, Se, Al, and Cu ranked second in concentration, surpassed only by the muscle (Table 2, Figure 1).

The elements arsenic (As), selenium (Se), zinc (Zn), and iron (Fe) exhibited highly notable correlations that were of the utmost importance and relevance. The interrelationships and connections between these elements were particularly striking and noteworthy, demonstrating their strong association and mutual influence.

Positive correlations of the arsenic contents between individual tissues were found between As in the spinous process and facet (0.8, *p* < 0.01) and between As in the paraspinal muscle and IVD (0.65, *p* = 0.02) and in the spinous process (0.6, *p* = 0.03). Moreover, As in the muscle significantly negatively correlated with Al (−0.56, *p* = 0.04). In the facet, As correlated significantly with Cu (0.62, *p* = 0.02) and Se (0.8, *p* < 0.01), and it negatively with Zn (−0.64, *p* = 0.02) and Se in the spinous process (0.58, *p* = 0.03) (Figure 2).

Se in the IVD correlated negatively with Al (−0.64, *p* = 0.02), Zn (−0.62, *p* = 0.02), and Fe (−0.58, *p* = 0.02), and it positively with Cu (0.71, *p* < 0.01) and Mo in the spinous process (−0.64, *p* = 0.02). In the IVD, Fe positively correlated with Cu (0.74, *p* < 0.01). In the spinous process, Se correlated with Cu (0.77, *p* < 0.01) (Figure 2).

Zn significantly negatively correlated with Se in the articular process (−0.56, *p* = 0.04) and in the spinous process with Se (−0.61, *p* = 0.02) and Zn (0.62, *p* = 0.02). Zn in the IVD negatively correlated with Al in muscle (−0.56, *p* = 0.04) and Cu in muscle (−0.56, *p* = 0.04). Zn in the spinous process positively correlated with Al (0.57, *p* = 0.04) *p* = 0.04). Zn in muscle positively correlated with Mo (0.67, *p* = 0.01) and negatively with Fe (−0.72, *p* < 0.01) in the facet (Figure 2).

Fe in the IVD correlated positively with Zn (0.74, *p* < 0.01) and negatively with Se (−0.58, *p* = 0.03) and Cu in muscle (−0.58, *p* = 0.03). Fe in muscle correlated negatively with Cu in the IVD (−0.55, *p* = 0.04) and positively with Mo (0.58, *p* = 0.03). Fe in the facet correlated with Se (0.68, *p* = 0.01) (Figure 2).

A negative correlation was found between arsenic (As) in the spinous process and copper (Cu) (−0.7, *p* = 0.02) and selenium (Se) (−0.6, *p* = 0.04) in the blood. Additionally, arsenic (As) in the blood was significantly negatively correlated with copper (Cu) in the muscle (−0.7, *p* = 0.03) and molybdenum (Mo) in the spinous process (−0.6, *p* = 0.04) (Figure 3).

Copper (Cu) in the spinous process was significantly negatively correlated with selenium (Se) in the blood (0.6, *p* = 0.03), and copper (Cu) in the intervertebral disc (IVD) was negatively correlated with molybdenum (Mo) in the blood (−0.7, *p* = 0.02) (Figure 3).

Zinc (Zn) in the blood was significantly correlated with molybdenum (Mo) in the muscle (0.8) and with iron (Fe) in the spinous process (0.7), and it was negatively correlated with molybdenum (Mo) in the facet (−0.6) and zinc (Zn) in the muscle (−0.7).

The pain duration was significantly correlated with aluminum (Al) in the intervertebral disc (IVD) (−0.7, *p* = 0.02). Pain on the Visual Analog Scale (VAS) was strongly and significantly correlated with arsenic (As) in the blood (0.9) and negatively with molybdenum (Mo) in the spinous process (−0.6, *p* = 0.04). The Comi Back Scale was positively correlated with molybdenum (Mo) in the IVD (0.7, *p* = 0.02) (Figure 3).

The hemoglobin (HGB) and red blood cell count (RBC) values positively correlated with selenium (Se) in the facet and spinous process (Figure 3).

The PCA analysis within individual tissues revealed similarities in the distribution of elements between the spinous process and facet. The difference between the tissues concerned Mo and As, indicating positive values for component 2 in the facet and for component 1 and component 2 in the spinous process, respectively (Figure 4).

For the muscle, similarities were observed for Zn and Se for component 2 and for Al, Cu, Se, and Mo for component 1. However, in the IVD, the elements Cu, As, Fe, and Se exhibited similarities for component 1.

The inner ellipses represent a 95% confidence level for each mean and the outer ellipses represent a 50% contour for each group, which is a region in the space of the two canonical variables that contains approximately 50% of the observations, assuming normality. The direction of a ray indicates the degree of association for a covariate with the two canonical variables. The coordinate points (canonical 1 and 2) correspond to group means of trace elements and are denoted by plus (“+”) signs (Table 3).

The table lists the eigenvalues, contributions, cumulative contributions, and canonical correlations obtained from the analysis. The eigenvalues for functions 1 and 2 were 2.59 (contribution: 65.1%) and 1.37 (contribution: 34.3%), respectively, showing that the function 1 variables contained significantly more information that can be used to distinguish the groups. By substituting the measurement data for each sample in the discriminant functions, we calculated score points for the function 1 and 2 variables. Based on these score points, we created a scatter diagram with the function 1 and 2 score points, plotted on the horizontal (canonical 1) and vertical axes (canonical 2), respectively (Figure 5).

The largest difference between tissues was observed along the first canonical axis (Figure 5). This dimension clearly separates the tissues into three distinct groups: the facet and spinous tissues are clearly distant from the muscle tissue, which is associated with higher levels of copper (Cu) and iron (Fe) in the intervertebral disc (IVD).The positions of the muscle along canonical 1 can be attributed to the higher concentrations of Fe, Cn, and Cu (Figure 5). 

The muscle tissue clusters separately from the spinous and facet tissues and is more strongly associated with elevated levels of As in osteophytes lower grade group (Appendix A). The muscle tissue clusters separately from the spinous and facet tissues and is more strongly associated with elevated levels of Fe in the Baastrup group (Appendix A). In the lower-grade Goutallier degeneration group muscle tissues shift closer to Al, suggesting a different elemental environment in advanced muscle degeneration (Appendix A). In Modic type 2 with spinous processes and muscle tissues aligning along vectors representing elevated levels of Se, Cu, and Mo, indicating altered metal homeostasis in this degeneration stage. The intervertebral disc is distinctly separated and projects along the aluminum (Al) axis, suggesting increased Al content in disc tissue at this stage. In Modic type 3 shows clearer tissue clustering, especially between the spinous and facet tissues, which are closely associated with Zn, Fe, and As (Appendix A). In the lower-grade Pfirmann group muscle tissues shift closer to Al (Appendix A).

## 4. Discussion

Our findings showed that the highest Al concentrations are found in the paraspinal muscle, followed by the IVD, spinous process, and facet. The Cu levels are relatively low and consistent across these tissues. The As levels are highest in the paraspinal muscle and IVD, while the facet and spinous process have lower concentrations. The Mo concentrations are generally low. The Se levels show moderate variability across tissues. The Zn concentrations are highest in the paraspinal muscle. The Fe concentrations are highest in the paraspinal muscle, followed by the IVD, facet, and spinous process. The paraspinal muscle exhibits the highest median values for trace metals.

Comparing the contents of the elements in the IVD with previous research [9], the Al content in the IVD shows a mean of 2.33 ± 2.49 and a median of 1.48 (0.51–2.76), while a lower mean of 0.63 ± 0.29 and a median of 0.55 (0.17–1.27) were reported. The Cu content in the IVD has a mean of 2.92 ± 1.22 and a median of 2.95 (1.99–3.31), similar to a previous study showing a mean of 2.5 (0.97–23.64) and median of 3.41 ± 4.05. The Mo content in the IVD has a mean of 0.07 ± 0.19 and a median of 0.02 (0.01–0.02), comparable to a previous study showing a mean of 0.05 ± 0.03 and median of 0.05 (0.02–0.14). The Se concentration in the IVD has a mean of 0.33 ± 0.17 and a median of 0.27 (0.24–0.36), while the previous study reported a higher mean of 0.46 ± 0.14. The Zn content in the IVD has a mean of 39.32 ± 21.1 and a median of 31.62 (22.49–58.31), whereas the previous study found a higher mean of 113.83 ± 10.11. Finally, the Fe content in the IVD has a mean of 292.16 ± 107.53 and a median of 269.9 (207.1–303.7), slightly lower than previous study’s mean of 340.35 ± 47.69.

Staszkiewicz et al. conducted a study where they compared degenerative intervertebral discs (IVDs) with a control group. Their findings revealed a significant relationship between the levels of copper (Cu) and iron (Fe) in the IVD and the extent of radiological changes according to the Pfirrmann scale. Furthermore, they identified other important connections in degenerate IVDs, particularly the correlations between Cu and Fe, as well as between Zn and Fe [22]. 

Nevertheless, it should be noted that in our earlier study, we also showed that the Fe concentrations were higher in IVDs obtained from the study group compared to controls but decreased as the severity of radiographic degeneration of the IVDs increased [22].

Furthermore, the research conducted by Staszkiewicz has provided valuable insights into the potential diagnostic significance of assessing copper (Cu) and zinc (Zn) levels in the bloodstream for predicting the occurrence of intervertebral disc degeneration (IVDD) in the lumbosacral region. The study findings indicated a notable and statistically significant reduction in the concentrations of Cu and Zn in the sera of individuals with IVDD when compared to the control group. Additionally, a strong correlation between Cu and Zn was observed, highlighting their interdependency. Moreover, the concentrations of Cu, Zn, calcium (Ca), and magnesium (Mg) were found to exhibit variations based on the extent of radiological changes evaluated using the Pfirrmann scale [23].

In the research conducted by Jakoniuk et al., they found that individuals diagnosed with osteoarthritis had lower levels of zinc (Zn) and copper (Cu) in their serum compared to the control group. Specifically, the osteoarthritis group exhibited average serum Zn and Cu levels of 0.77 ± 0.22 mg/L and 1.1 ± 0.35 mg/L, respectively, whereas the control group had levels of 0.83 ± 0.13 mg/L for Zn and 1.25 ± 0.41 mg/L for Cu. Additionally, within the study group, it was observed that smokers had significantly lower Cu concentrations (1.07 ± 0.35 mg/L) compared to non-smokers (1.17 ± 0.34 mg/L). These findings suggest that there may be a relationship between osteoarthritis and altered levels of Zn and Cu in the blood [24].

Hakan et al. conducted a study where they examined the levels of aluminum (Al) and selenium (Se) in different parts of the body, such as the serum, bone, and intervertebral disc (IVD), in patients with lumbar herniated nucleus and lumbar stenosis [25]. They showed that the Al disc level in the HNP group was found to be significantly higher than the Al disc level in the LSS group (*p* = 0.038). While the As serum level increased in LDH group, it was determined that the As bone level increased very significantly (r = 0.699, *p* = 0.001). In our study, we showed high correlations of As contents in individual tissues. This suggests that Al may play a role in the development or progression of these conditions. Additionally, the significant increase in arsenic (As) levels in both serum and bone samples in the LDH group raises questions about its potential impact on the disease. One possibility is that the accumulation of Al in the intervertebral disc (IVD) contributes to the pathogenesis of HNP, possibly by promoting degeneration or inflammation in the disc. Similarly, the elevated As levels in serum and bone samples may be associated with disease processes, although the exact mechanisms remain unclear. Future research should explore how these trace elements interact with the surrounding tissues and cells, as well as investigate the impact of long-term exposure to Al and As on disc health and spinal conditions.

The observed negative correlation between Iron (Fe) and copper (Cu) in the paraspinal muscle and intervertebral discs might be linked to their roles in various metabolic pathways. One study highlighted that copper-dependent enzymes, crucial for antioxidant defenses, mitochondrial energy production, and iron metabolism, are affected in the muscles of patients with profound copper deficiency, leading to myeloneuropathy [26]. This suggests that imbalances in copper can significantly impact iron metabolism, with potential effects on muscle and spinal health. The study also observed changes in the expression of copper and iron-related proteins in muscles under copper deficiency conditions.

The observed positive correlations of 0.6 between iron (Fe) and molybdenum (Mo) and 0.52 between molybdenum (Mo) and copper (Cu) in paraspinal muscles suggest intricate molecular interactions and functional dependencies among these trace elements that are critical for muscle physiology. Fe, Mo, and Cu are integral components of various enzymes involved in the electron transport chain within mitochondria. Iron is a key component of cytochromes and iron–sulfur (Fe-S) clusters that facilitate electron transfer. Molybdenum is essential for the activity of certain oxidases that interact with the ETC indirectly. Copper is a critical component of cytochrome c oxidase (complex IV), which accepts electrons from cytochrome c. The correlation could reflect a coordinated demand for these metals to maintain efficient ATP production in energy-demanding paraspinal muscles. This is supported by studies into mitochondrial efficiency and the role of trace metals in bioenergetics [27].

The correlations between copper (Cu), zinc (Zn), and iron (Fe) in the bone tissue of the facet and spinous processes suggest complex interactions and essential roles of these trace elements in bone metabolism and structural integrity. The dynamic process of bone remodeling, involving bone resorption by osteoclasts and formation by osteoblasts, requires the coordinated action of Cu, Zn, and Fe. Zinc stimulates osteoblastic bone formation and inhibits osteoclastic bone resorption, thereby playing a critical role in the maintenance of bone mass. Copper influences the activity of lysyl oxidase, an enzyme necessary for the maturation of collagen in the bone matrix. Iron is vital for the proliferation and differentiation of osteoblasts, partly through its role in DNA synthesis. The interactions among these metals can significantly impact bone remodeling and repair processes, as well as the role of zinc in bone metabolism [28].

Positive correlations of arsenic contents between individual tissues were found between As in the spinous process and facet and between As in the paraspinal muscle and IVD and in the spinous process. As is known to enter the body primarily through ingestion and inhalation. After absorption, arsenic is distributed throughout the body via the bloodstream. It undergoes biomethylation, primarily in the liver, where inorganic arsenic is metabolized to less toxic methylated metabolites. This process of methylation is considered a detoxification mechanism [29].

The correlations observed between different elements and compounds in the tissues could be influenced by several factors, including the distribution and metabolism of arsenic in the body, as well as the interactions between different elements [30]. The positive correlations between As in different tissues suggest a common source or exposure to arsenic. The negative correlation between As in paraspinal muscle and Al could indicate a potential antagonistic relationship between these elements.

The negative correlations observed between Zn and Se in different tissues could suggest a competitive interaction or regulation between these elements. Zn and Se are vital trace elements with roles in biological processes. They compete for binding sites on proteins, enzymes, and transcription factors. For instance, zinc finger proteins and selenoproteins require either zinc or selenium for proper structure and function. The increased availability of one element can outcompete the other, leading to a negative correlation between their levels. In terms of antagonistic redox roles, zinc and selenium contribute to antioxidant defense. Selenium is crucial for glutathione peroxidase (GPX), a key antioxidant enzyme, while zinc regulates metallothioneins (MT) involved in detoxification. Their interplay in redox regulation and opposing roles in antioxidant pathways may contribute to the observed negative correlation. In terms of the nutritional status and compensatory response, aging affects zinc and selenium levels, increasing deficiency risks. The negative correlation may result from a compensatory response to counteract age-related damage. The upregulation of zinc- and selenium-dependent antioxidant proteins could reflect self-induced oxidative stress [31].

A notable finding was the negative correlation between arsenic (As) in the spinous process and copper (Cu) and selenium (Se) in the blood. This finding is consistent with previous research indicating that arsenic, a toxic metalloid, can disrupt the normal metabolic functions of other trace elements. The negative association between As in the spinous process and Se in the blood may point to a complex interplay where arsenic toxicity impairs the physiological roles of selenium, an essential antioxidant that supports cellular function and inflammation regulation [32].

Similarly, the significant negative correlation between As in the blood and Cu in the muscle and Mo in the spinous process further supports the hypothesis that arsenic might interfere with other trace elements that are crucial for tissue repair and function. Copper, in particular, is vital for collagen synthesis and the integrity of connective tissue, while molybdenum plays an essential role in enzymatic functions in tissues, including the spine. The disruption of these elements by arsenic could impair the regenerative processes of spinal tissues, contributing to pain and dysfunction [33].

Pain duration was significantly correlated with aluminum (Al) in the intervertebral disc (IVD), which aligns with research indicating that aluminum exposure can influence bone and cartilage integrity, potentially leading to disc degeneration. Moreover, pain intensity in the VAS was strongly correlated with arsenic (As) in the blood, further supporting the notion that toxic elements such as arsenic could exacerbate pain perception, possibly through inflammatory pathways [34].

The positive correlation between molybdenum (Mo) in the IVD and the Comi Back Score suggests that adequate levels of molybdenum in the intervertebral disc may be beneficial in improving spinal function and reducing discomfort, potentially through the involvement in enzymatic processes and tissue repair mechanisms [7]. This study has several limitations that should be acknowledged. First, the small sample size (n = 13) limits the generalizability of the findings. Although the study provides novel insights into the role of trace elements in degenerative spine tissues, a larger cohort would be necessary to draw more definitive conclusions. Second, the study’s observational design does not allow for establishing causality between trace element concentrations and degenerative spine disease. The cross-sectional nature of the research precludes tracking changes over time, which would be valuable in understanding disease progression. These factors could significantly affect trace element concentrations in tissues and might influence the observed correlations. Future studies should incorporate comprehensive patient histories, including dietary habits and lifestyle factors, to better account for these variables. Moreover, while we utilized multiple grading scales (Modic, Pfirrmann, Goutallier, and Pathria scales) to assess spinal degeneration, other clinical parameters such as pain intensity, functional disability, and biochemical markers were not included. Integrating these aspects into future studies would provide a more comprehensive understanding of the relationship between trace element concentrations and spinal degeneration. Despite these limitations, our study contributes to the growing body of research on metabolic- and oxidative-stress-related factors in degenerative spine disease, highlighting the need for further investigation with expanded sample sizes and more controlled study designs.

It is important to note that the analyzed group in the mentioned study consisted of patients qualified for surgical treatment; therefore, the findings can only partially represent the general group of patients with degenerative changes of the spine. Further studies with larger sample sizes and broader representation would be necessary to generalize the findings to a wider population.

In summary, the quantitative analysis of trace elements in degenerative tissues provides valuable information about metabolic processes, oxidative stress, and pathological changes. Essential elements such as Cu and Zn play crucial roles in antioxidant defense mechanisms, enzymatic activity, and inflammatory regulation. Instrumental techniques, such as AAS, ICP-OES, and ICP-MS, enable the analysis of trace elements in tissues, while validation ensures the reliability of the results. Further research is necessary to elucidate the interplay between trace elements, oxidative stress, and inflammatory responses in degenerative processes, particularly in spinal pathologies, which could provide new insights into potential therapeutic strategies.

The analyzed group in this study comprised patients who were selected for surgical treatment. Consequently, the characteristics and findings of this group may only partially represent the broader population of individuals with degenerative changes in the spine. The specific inclusion criteria and treatment selection process may introduce certain biases and limit the generalizability of the results to the wider population.

## 5. Conclusions

The study revealed significant variations in trace element concentrations across spinal tissues in patients with degenerative changes, emphasizing their potential role in oxidative stress, inflammation, and metabolic dysregulation. Notably, aluminum (Al), copper (Cu), and iron (Fe) exhibited the highest concentrations in paraspinal muscles, suggesting their involvement in redox homeostasis and degenerative processes. The observed correlations between these elements across tissues highlight a complex interplay influencing spinal tissue metabolism and degeneration. Arsenic (As) showed strong positive correlations between the spinous process and facet, and between the paraspinal muscle and intervertebral disc (IVD), suggesting arsenic’s systemic distribution and its potential involvement in degenerative spine conditions. The significant negative correlation of As with Al in muscle tissue and its positive correlations with Cu and selenium (Se) in the facet suggest intricate interactions that may impact oxidative stress pathways, inflammation, and tissue repair mechanisms. Selenium (Se) displayed negative correlations with Al, Zn, and Fe in the IVD, and positive correlations with Cu, indicating Se’s critical role in antioxidant protection and its interactions with other trace elements that may affect oxidative stress and inflammation pathways in degenerative spinal tissues. The correlations of zinc (Zn) suggest its involvement in tissue repair and inflammatory responses, indicated by negative correlations with Se and positive correlations with molybdenum (Mo) in muscle tissue. Iron (Fe) exhibited positive correlations with Zn in the IVD and negative correlations with Se and Cu in muscle tissue, reflecting its essential role in oxygen transport, energy metabolism, and possibly in oxidative stress regulation. Given Fe’s dual role in redox reactions and inflammatory pathways, its altered levels may contribute to pathological oxidative imbalances in degenerative spine disorders. These findings contribute to understanding the pathophysiology of DLSD and underscore the potential of trace element analyses in diagnosing and developing therapeutic strategies for spinal degenerative diseases. The study revealed significant negative correlations between arsenic (As) levels and essential trace elements such as copper (Cu) and selenium (Se), suggesting a potential antagonistic relationship impacting tissue mineral balance. The pain intensity and duration were significantly associated with specific metal concentrations, indicating that elements such as arsenic (As), aluminum (Al), and molybdenum (Mo) may play a role in the pathophysiology of spinal pain. Further research involving larger populations is warranted to confirm these associations and explore the diagnostic and therapeutic implications of trace element imbalances in spinal degeneration, particularly in the context of redox homeostasis and antioxidant defense mechanisms.

## Figures and Tables

**Figure 1 antioxidants-14-00485-f001:**
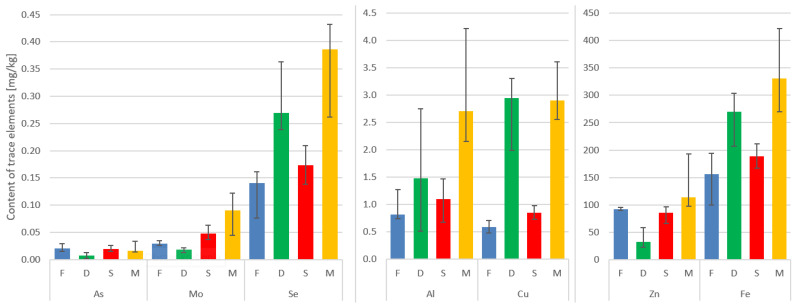
Median concentrations and variability of trace elements in the four tissues. Vertical lines on top of the bars represent the Q1–Q3 range of the median.

**Figure 2 antioxidants-14-00485-f002:**
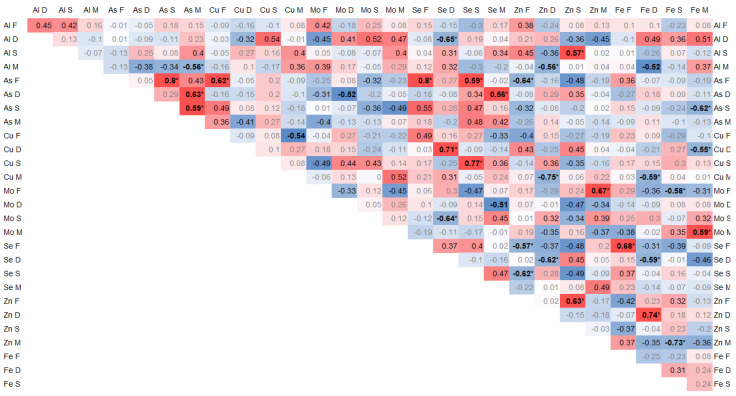
Spearman correlation trace elements in 4 tissues, where the color intensity represents the strength and direction of the correlation coefficients, ranging from −1 (strong negative correlation, dark blue) to +1 (strong positive correlation, dark red); * statistically significant correlations.

**Figure 3 antioxidants-14-00485-f003:**
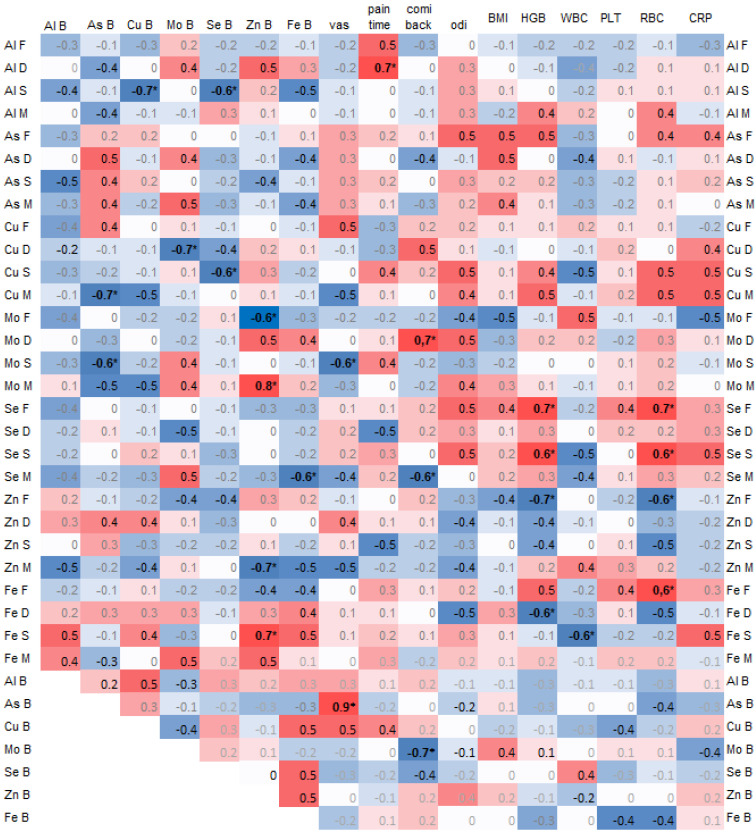
Spearman correlations of trace elements in 4 spine tissues, namely the paraspinal muscle (M), IVD (D), facet (F), and spinous process (S), and with trace elements in blood (B) and clinical data, where the color intensity represents the strength and direction of the correlation coefficients, ranging from –1 (strong negative correlation, dark blue) to +1 (strong positive correlation, dark red); * statistically significant correlations.

**Figure 4 antioxidants-14-00485-f004:**
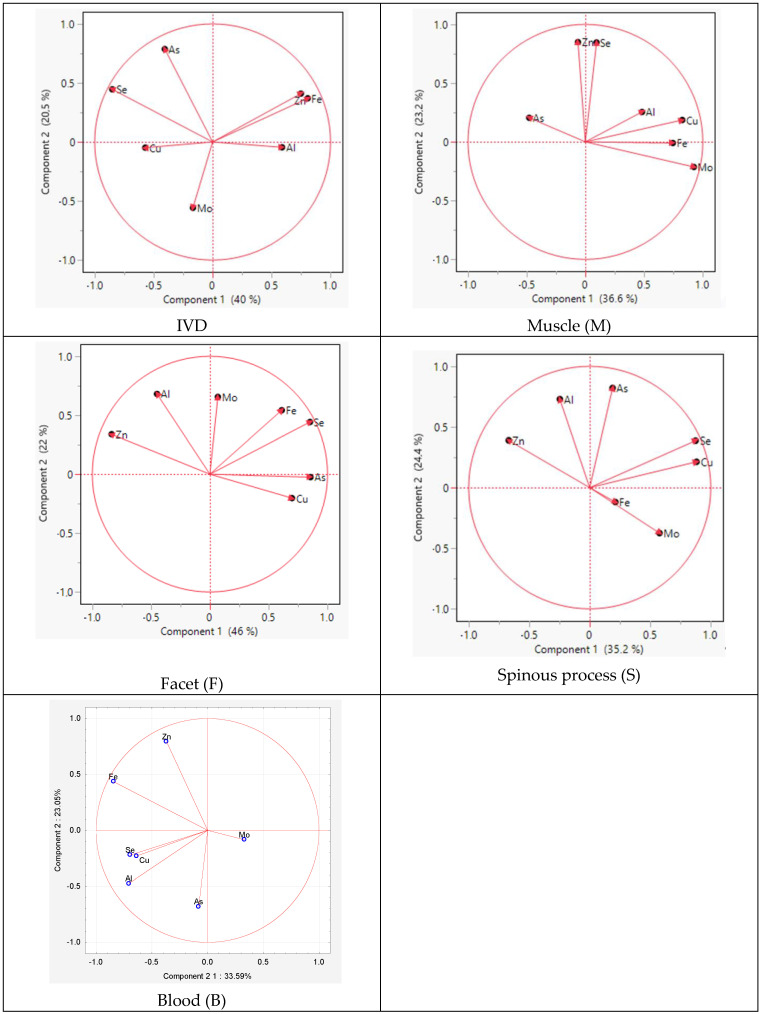
Graphic illustration of the principal component analysis of the trace element content. Projection of the variables on the factor plane of the first two principal factors in tissues.

**Figure 5 antioxidants-14-00485-f005:**
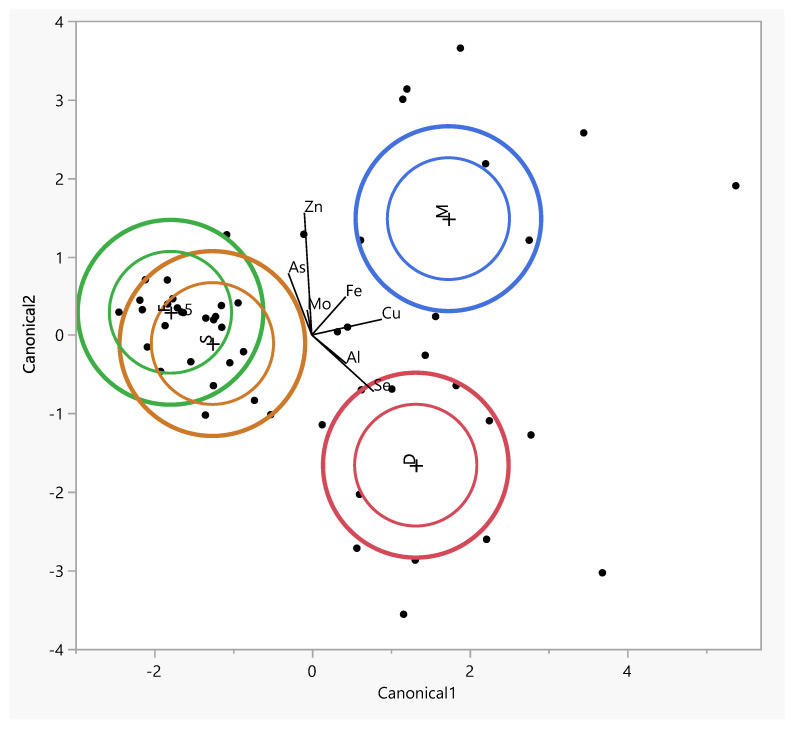
The canonical discriminant analysis showing interspecific variations in the concentrations of trace elements in muscle (M) and facet (F), spinous (S), and intervertebral disk (D) tissues. Ellipses represent the 95% confidence intervals around the group centroid of each tissue.

**Table 1 antioxidants-14-00485-t001:** Information on patients included in the study (*n* = 13).

Parameters	Values
Age (years)	58.77 ± 10.48
Sex (Male/Female)	10/3
BMI (kg/m^2^)	25.3 ± 4.4
Facet joint osteophytes (2/3/4)	5/6/2
Pathria scale (2/3)	8/5
Goutallier scale (1/2/3)	1/7/5
Modic scale (2/3)	5/8
Pfirrmann scale (2/3/4/5)	3/4/3/3
BMI	26.39 ± 4.36
VAS	7.1 ± 1.5
Duration of pain symptoms (years)	4.23 ± 2.26
ODI	58.69 ± 10.95
Comi Back	7.72 ± 1.2
Laboratory Test	
HGB (hemoglobin) (g/dL)	12.08 ± 2.55
WBC (white blood cells) (×10^3^/µL)	8.38 ± 1.86
PLT (platelets) (×10^3^/µL)	227.23 ± 77.7
RBC (red blood cells) (×10^6^/µL)	3.96 ± 0.89
CRP (C-reactive protein) (mg/L)	10.71 ± 7.12
Comorbidities:	
Arterial hypertension	9
Type 2 diabetes	5
Anxiety neurosis	2
Coronary heart disease	2
Asthma	1
Chronic kidney disease	1

**Table 2 antioxidants-14-00485-t002:** Concentrations of trace elements (Al, As, Mo, Zn, Cu, Fe, Se) in spinal tissues: paraspinal muscle (M), IVD (D), facet (F), spinous process(S) (in mg/kg on dry mass basis), and blood (B) (in mg/L). SD—standard deviation; IQR—interquartile range.

Trace Elements	Tissue	Mean ± SD	Median (IQR)
Al	Facet	1.03 ± 0.72	0.82 (0.74–1.27)
	IVD	2.33 ± 2.49	1.48 (0.51–2.76)
	S	1.16 ± 0.73	1.1 (0.68–1.47)
	M	3.35 ± 3.09	2.71 (2.16–4.22)
	B	0.06 ± 0.1	0 (0–0.11)
As	F	0.02 ± 0.01	0.02 (0.02–0.03)
	D	0.02 ± 0.03	0.01 (0–0.01)
	S	0.02 ± 0.01	0.02 (0.02–0.03)
	M	0.03 ± 0.02	0.02 (0.01–0.03)
	B	0.01 ± 0.01	0.01 (0.01–0.01)
Cu	F	0.6 ± 0.14	0.59 (0.49–0.71)
	D	2.92 ± 1.22	2.95 (1.99–3.31)
	S	0.86 ± 0.26	0.85 (0.73–0.98)
	M	3.15 ± 1.29	2.9 (2.55–3.61)
	B	0.97 ± 0.28	0.9 (0.86–1.03)
Mo	F	0.04 ± 0.02	0.03 (0.03–0.03)
	D	0.07 ± 0.19	0.02 (0.01–0.02)
	S	0.06 ± 0.03	0.05 (0.04–0.06)
	M	0.1 ± 0.07	0.09 (0.04–0.12)
	B	<LOD	0 (0–0.01)
Se	F	0.13 ± 0.06	0.14 (0.08–0.16)
	D	0.33 ± 0.17	0.27 (0.24–0.36)
	S	0.18 ± 0.05	0.17 (0.14–0.21)
	M	0.36 ± 0.13	0.39 (0.26–0.43)
	B	0.09 ± 0.02	0.08 (0.08–0.1)
Zn	F	92.18 ± 14.58	91.41 (89.36–95.24)
	D	39.32 ± 21.1	31.62 (22.49–58.31)
	S	80.35 ± 20.85	85.12 (66.07–96.53)
	M	133.52 ± 63.1	114.1 (97.08–193.3)
	B	6.17 ± 2.47	5.62 (4.82–6.67)
Fe	F	153.03 ± 72.2	155.7 (99.45–194.5)
	D	292.16 ± 107.53	269.9 (207.1–303.7)
	S	192.56 ± 68.93	188.2 (167.2–211.3)
	M	406.88 ± 208.56	330.4 (269.9–422.7)
	B	482.77 ± 60.43	463.2 (444.6–497.4)

**Table 3 antioxidants-14-00485-t003:** Eigenvalues, percentages of variance explained, and standardized canonical discriminant functions for concentrations of trace elements in muscle and facet, spinous, and intervertebral disk tissues. *—statistically significant canonical functions.

Eigenvalue	Percent	Cum Percent	Canonical Corr	Likelihood Ratio	Approx. F	NumDF	DenDF	Prob > F
2.59	65.1	65.1	0.85	0.12	6.49	21	121.15	<0.0001 *
1.37	34.3	99.4	0.76	0.41	3.99	12	86	<0.0001 *

## Data Availability

The data presented in this study are available on request from the corresponding author.

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
