# Peer review of "Trace Element Concentrations in Degenerative Lumbar Spine Tissues: Insights into Oxidative Stress"

_antioxidants, 2025, doi:10.3390/antiox14040485_

Round 1

Reviewer 1 Report

The work I reviewed demonstrates great scientific value. The studies of trace amounts of elements were conducted reliably and with appropriate statistical analysis.

The authors should note certain shortcomings of the work:

1. page 2 lines 77 and 79 - information is duplicated.

2. page 4 line 134 - it is IPC-MS and should be ICP-MS.

3. page 4 line 148 - it is concentrations and should be concentration.

4. page 5 line 207 - In my opinion, the title of the chapter is completely inappropriate, and the sentence in line 208 is unnecessary.

5. I lacked a deeper discussion based on the research of other scientists.

Author Response

Dear Reviewer,

Thank you for your valuable comments and suggestions, which have helped us improve our manuscript. Below, we provide our responses to each of your remarks:

  1. page 2 lines 77 and 79 - information is duplicated.
    – We appreciate your observation regarding the duplicated information. We have revised the text to remove redundancy.

  2. page 4 line 134 - it is IPC-MS and should be ICP-MS.
    – Thank you for pointing out the typographical error. We have corrected "IPC-MS" to "ICP-MS."

  3. page 4 line 148 - it is concentrations and should be concentration.
    – We have corrected "concentrations" to "concentration" as suggested.

  4. page 5 line 207 - In my opinion, the title of the chapter is completely inappropriate, and the sentence in line 208 is unnecessary.
    – We have removed the unnecessary sentence.

  5.  I lacked a deeper discussion based on the research of other scientists.
    – We appreciate your suggestion to include a broader discussion. We have expanded this section by incorporating additional references and insights from relevant studies.

We are grateful for your thoughtful review, which has contributed to improving the clarity and quality of our manuscript.

Best regards,

Mikolaj Dabrowski

Reviewer 2 Report

The authors of the manuscript ID: 3522676 entitled,, Exploring Trace Element Content in Degenerative Spine Tissues” submitted the manuscript for review as a potential publication in the journal Antioxidants.  

The aim of the study was to investigate the concentrations of trace elements, i.e. zinc , arsenic , copper, selenium, iron, magnesium, and potassium in the spine tissues (intervertebral discs, muscles and bones) in patients with degenerative lumbar spine disease and their potential associations with the disease.

The research involved 13 patients (3 women and 10 men), undergoing surgery for symptomatic degenerative spine disease. The tissues samples were collected during interbody fusion of the spine and perpendicular stabilization of the lumbar spine. Trace element concentrations were analyzed using chemical and radiographic assessments.

Results showed significant variations and correlations among trace elements across different spinal tissues. The authors suggest their roles in metabolic, oxidative processes and pathology of spinal degeneration.

However, there is no solid clinical evidence to support the conclusions suggested by the authors.The main problem is that the authors did not provide and did not correlate the concentrations of trace elements with clinical parameters. In order for the manuscript to be useful, the authors should supplement it with clinical data of the patients and then demonstrate a potential relationship or not with the elements studied. Te article must be corrected.

In my opinion, the patient data is incomplete, which makes the results of little use. Since there are no reference values ​​for the given elements in tissues, it is difficult to speculate on what significance they have in patients in the course of spine degeneration. Correlations of one element with another do not contribute much to the clinical assessment of pain and spinal degeneration. I assume that before the surgery, the patients had a blood count, perhaps iron, hemoglobin, c-reactive protein or lipid levels. Do the detected quantitative concentrations of elements have any relationship with clinical parameters? There is also no basic clinical information about other diseases of the patients and how long they have had pain. In order for the manuscript to be useful, the authors should supplement it with clinical data of the patients and then demonstrate a potential relationship with the elements studied. The most interesting element of the manuscript is the graphical presentation (4 figures) of the obtained results. The statistical analysis used provided an expansion of the formula for incomplete clinical data. I think that this will be an excellent article after the changes applied.

Author Response

Dear Reviewer,

Thank you for your thorough review and insightful comments on our manuscript. We appreciate your time and effort in evaluating our work. Below, we address your concerns and describe the revisions made to improve the manuscript.

  1. Lack of clinical parameters and their correlation with trace element concentrations
    We acknowledge the importance of integrating clinical parameters to strengthen the interpretation of our findings. In response to your comments, we have supplemented the manuscript with additional clinical data, including the duration of spinal pain, previous treatments, and comorbidities of the studied patients. Furthermore, we have reanalyzed the data to assess potential correlations between trace element concentrations and these clinical parameters. The results of this analysis are now included in the revised manuscript.

  2. Clinical significance of trace element concentrations
    We understand the challenges of interpreting trace element levels without established reference values in spinal tissues. To address this, we have expanded the discussion by incorporating comparisons with existing literature on trace elements in human tissues and their potential role in degenerative processes. We have also refined our conclusions to better reflect the exploratory nature of our study while emphasizing the need for further research in this area.

  3. Additional laboratory parameters (e.g., blood count, iron, hemoglobin, CRP, lipid levels)
    While our study primarily focused on trace elements in spinal tissues, we recognize the potential value of systemic biochemical parameters in understanding disease pathology. Unfortunately, not all patients had a full set of preoperative laboratory tests available. However, where possible, we have included relevant clinical laboratory data and analyzed their potential relationships with trace element concentrations.

  4. Manuscript revisions and improvements
    Based on your suggestions, we have revised the manuscript to provide a clearer and more clinically relevant interpretation of the results. We have also ensured that the graphical presentation remains a strong aspect of the manuscript, supporting the findings in a visually informative manner.

We sincerely appreciate your constructive feedback, which has significantly contributed to enhancing the quality of our work. We hope that the revisions now address your concerns and improve the overall clarity and impact of our study.

Best regards,

Mikolaj Dabrowski

Reviewer 3 Report

This study investigated the role of trace elements in degenerative lumbar spine disease, which contributes valuable insights into potential metabolic and oxidative stress-related pathways in spinal degeneration. The use of inductively coupled plasma mass spectrometry for trace element analysis is a precise technique. 

Minor comments:

Title: "Trace Element Concentrations in Degenerative Lumbar Spine Tissues: Insights into Oxidative Stress" is better?

Line 44-45: "influencing cellular homeostasis, inflammatory processes, and extracellular matrix degradation" is vague." Consider specifying mechanisms, "Oxidative stress disrupts cellular homeostasis by increasing reactive oxygen species (ROS), which trigger proinflammatory cytokine release and degrade extracellular matrix components like collagen."

Line 72: Fluoride (F) is listed as a trace element of interest in the study aim, but no results or discussion are provided. If fluoride was measured, include the data; if not, clarify why it was omitted despite being mentioned.

Line 77: only 13 patients, which is a limitation but not justified in the methods

Line 80: There is no mention of controlling for confounders such as diet, occupation, smoking status, or environmental exposures, which could influence trace element levels.

Table 1: the text could elaborate on how these characteristics relate to the study’s objectives

Figure 2: The Spearman correlation results are detailed but lack a summary of key findings. please highlight the most significant correlations.

Discussion:

Limitations: the small sample size and observational nature, the confounder control (diet, smoking) is not addressed. Expand this section to include these factors and their potential impact on results.

Line 343-348:This paragraph lacks references.

Author Response

Dear Reviewer,

We sincerely appreciate your constructive feedback and thoughtful suggestions, which have helped us refine our manuscript. Below, we address your comments and describe the revisions made to improve the manuscript.

Major Comments

  1. Title Suggestion
    Thank you for the suggestion. We have revised the title to "Trace Element Concentrations in Degenerative Lumbar Spine Tissues: Insights into Oxidative Stress" to better reflect the study's focus.

  2. Clarification of Cellular Mechanisms (Lines 44-45)
    We agree that the original statement was too vague. We have modified it to provide a more precise explanation:
    "Oxidative stress disrupts cellular homeostasis by increasing reactive oxygen species (ROS), which trigger proinflammatory cytokine release and degrade extracellular matrix components such as collagen."

  3. Fluoride (F) Mentioned but Not Discussed (Line 72)
    Thank you for catching this inconsistency. There was an editorial mistake in the manuscript—our study focused on measuring iron (Fe), not fluoride (F). We have corrected this error in the study aim to accurately reflect the elements analyzed.
  4. Justification of Small Sample Size (Line 77)
    We acknowledge that the limited sample size is a study limitation. We have now included a justification in the Limitation, explaining that the study is exploratory, and patient recruitment was constrained by the availability of surgical candidates and tissue sample collection feasibility.

  5. Control for Confounding Factors (Line 80)
    We recognize the importance of potential confounding factors such as diet, occupation, smoking status, and environmental exposures. A questionnaire interview was conducted regarding exposure to these factors. We have added this information in the methodology.

  6. Elaboration on Patient Characteristics in Table 1
    We have expanded the text accompanying Table 1 to explain how these characteristics relate to the study’s objectives and potential trace element variations.

  7. Summary of Key Findings in Figure 2
    We have revised the figure legend and Results section to highlight the most significant correlations and their potential implications.

  8. Expansion of the Limitations Section
    We have expanded the Limitations section to explicitly discuss the small sample size, the observational nature of the study,

  9. Missing References (Lines 343-348)
    We have added relevant references to support the statements in this paragraph.

Thank you again for your valuable feedback, which has helped us improve the clarity, accuracy, and impact of our manuscript. We hope that our revisions adequately address your concerns.

Best regards,

Mikolaj Dabrowski

Round 2

Reviewer 2 Report

  The authors provided clinical data and improved the manuscript. I thank them for their contribution to improving the article.

The authors provided clinical data and improved the manuscript. I thank them for their contribution to improving the article.